# Alpha-Methyldopa May Attenuate Insulin Demand in Women with Gestational Diabetes Treated with Betamethasone

**DOI:** 10.3390/healthcare10010135

**Published:** 2022-01-10

**Authors:** Ioannis Kakoulidis, Costas Thomopoulos, Ioannis Ilias, Stefanos Stergiotis, Stefanos Togias, Aikaterini Michou, Charalampos Milionis, Evangelia Venaki, Eftychia Koukkou

**Affiliations:** 1Department of Endocrinology, Diabetes and Metabolism, Elena Venizelou General and Maternity Hospital, GR-11521 Athens, Greece; i_kakoulidis@yahoo.gr (I.K.); stef_ster@hotmail.com (S.S.); s.tog90@gmail.com (S.T.); katerina.michoy@yahoo.com (A.M.); pesscharis@hotmail.com (C.M.); evivenaki@gmail.com (E.V.); ekoukkou@gmail.com (E.K.); 2Department of Cardiology, Elena Venizelou General and Maternity Hospital, GR-11522 Athens, Greece; thokos@otenet.gr

**Keywords:** gestational diabetes, hypertension, pregnancy-induced, methyldopa, insulin, human

## Abstract

Gestational diabetes mellitus (GDM) is associated with hypertensive disorders in pregnancy. Alpha-methyl-DOPA (αMD) is a commonly used medication for hypertension in pregnant women. This medication may be associated with alteration in insulin resistance and glucose homeostasis. The aim of the present study was to investigate in 152 pregnant women whether the demands of exogenous insulin in glucocorticoid-treated women during pregnancy are different between those with GDM and hypertension treated with αMD and those without hypertension. In the group of women with GDM under insulin treatment, who received αMD for hypertension, the increase in insulin needs was relatively lower by at least 30% of the pre-admission insulin dose compared to all of the remaining women not receiving αMD in the same group (9 women vs. 50 women, *p* = 0.035). Our work raises the hypothesis that αMD can favorably modulate insulin sensitivity in the third trimester of pregnancy in previously insulin-treated women with gestational diabetes who receive glucocorticoids.

## 1. Introduction

Gestational diabetes mellitus is associated with hypertensive disorders in pregnancy [1,2]. Alpha-methyl-DOPA (αMD) may be used alone or combined with other drugs such as labetalol or nifedipine to treat increased blood pressure levels in pregnancy [1]. Based on an earlier report, it has been hypothesized that αMD may increase insulin resistance [3], but current empirical evidence denies any clinically relevant glycemic change following αMD treatment [4]. During the third trimester, in order to accelerate fetal lung maturation, glucocorticoids (GCs) are usually administered in high-risk pregnancies. However, GCs steadily increase plasma glucose levels making the administration of exogenous insulin in these GCs-treated women largely unavoidable [5,6,7].

The present investigation aimed to test whether the demands of exogenous insulin in GCs-treated women during pregnancy are different between those with gestational diabetes mellitus treated with αMD and women without hypertension.

## 2. Subjects/Methods

### 2.1. Setting

We retrospectively examined all the medical files of women with a singleton pregnancy admitted between August 2016 and the end of December 2019 to the Department of Obstetrics of our hospital during the third trimester of pregnancy. The indication for admission was a high-risk pregnancy because of obstetric complications (e.g., hydramnios, premature rupture of membranes, uterine contractions, vaginal bleeding). Women received betamethasone 12 mg daily for two days, according to our hospital’s protocol for high-risk pregnancies.

### 2.2. Subjects

Among women with a singleton pregnancy hospitalized for maternal complications during the third trimester (*n* = 450), we excluded high-risk pregnancies admitted for medical complications like uncontrolled gestational diabetes mellitus, uncontrolled hypertension, febrile conditions, or other systemic illness needing hospital admission (*n* = 170). We also excluded women in whose GCs administration was incomplete because of emergency delivery (*n* = 68), those being pregnant by assisted reproductive techniques (*n* = 15), women with gestational hypertension under treatment with drugs other than or in combination with αMD (*n* = 20), and those with an incomplete medical record for biochemical glucose metabolism assays (*n* = 25). We finally analyzed data from 152 women with a singleton pregnancy, hospitalized during the third trimester for obstetric complications, at high risk for premature delivery, and treated with CCs to accelerate fetal lung maturation. Women with already diagnosed gestational diabetes mellitus under ongoing insulin treatment and/or dietetic measures were not excluded. To exclude pre-existing gestational diabetes, all normoglycemic women at admission had a negative 75 g glucose tolerance test performed between the 24th and 28th week of gestation.

### 2.3. Insulin Treatment

All women received the pre-specified GCs dose upon admission and the following day. Glycemia assessment was made with six to seven capillary blood glucose measurements per day (pre-and one-hour post-prandial, plus an overnight measurement—whenever possible). Use of point of care devices (Exactive Vital; Enzyme GDH-FAD Fresh whole blood device; MicroTech Medical Campell, CA, USA) was implemented. All women were under a balanced dietary regimen during hospitalization. Insulin treatment was guided by obtained glycemic levels setting as a threshold of 90 mg/dL for fasting and 140 mg/dL for the one-hour post-prandial measure [8,9,10]. For women already on insulin treatment, we considered a cut-off of +30% in insulin dose for the analysis [11,12,13].

### 2.4. Statistical Analysis

Analyses were done with SPSS v.23.0 (IBM Corp., 2018, Armonk, NY, USA). Because of the absence of skewed distribution, continuous descriptive variables are reported as means and standard deviations, while categorical variables are reported as percentages. Significant differences between the study subgroups were determined using the chi-squared test. Analysis of increment in insulin dosing (in women with gestational diabetes already under insulin treatment) or of starting insulin (in those under dietetic measures or without gestational diabetes) versus αMD therapy was done with the “N-1” Chi-squared test as recommended by Campbell [14] and Richardson [15]. Statistical significance was set at *p* < 0.05. A post-hoc power calculation was also carried out.

## 3. Results

The mean age of the pregnancy cohort was 34 ± 5 years, while the mean gestational age was 32.7 ± 5.6 weeks. Mean weight gain during pregnancy was 8.7 ± 5.4 Kg. Fifty-nine women had gestational diabetes mellitus already treated with a median dose of 32 insulin units per day (group 1), 53 had gestational diabetes mellitus on medical nutrition therapy (group 2), and 40 women were without gestational diabetes mellitus (group 3). In our retrospective cohort, 17 (11.1%) women (age, 35 ± 5 years; gestational age, 32.6 ± 5.0 weeks; and weight gain, 7.9 ± 5.5 Kg) developed gestational hypertension and were treated with αMD at doses from 250 to 1000 mg/day. Women with gestational hypertension were more frequently represented in the two gestational diabetes subgroups (group 1, *n* = 9; and group 2, *n* = 6) than the non-gestational diabetes group (group 3, *n* = 2).

There was a considerable change in the glycemic profile of most women, mainly during the first 24 h of betamethasone administration, which lasted on average for 1.5 ± 1.0 days (group 1, 2.0 ± 0.9 days; group 2, 1.4 ± 1.2 days; and group 3, 1.0 ± 0.9 days). One hundred and twelve women needed either an increase of at least 30% in insulin dosing (group 1, *n* = 41 [69%]) or started insulin treatment (group 2 and 3, *n* = 71 [69%]) to maintain whole-day glycemia levels within target. More specifically, the mean insulin increment in group 1 was 10 units per day for three days, whereas the need for insulin in the remaining groups was 15 units for the same period (Table 1). In group 1, women treated with αMD required less frequently an insulin increase by at least 30% compared to the pre-admission insulin dose compared to all of the remaining women in the same group (*p* = 0.035). The post-hoc calculation was possible for group 1 only; with α = 0.10, the attained power was 60% [16].

## 4. Discussion

Our work raises the hypothesis that αMD can favorably modulate insulin sensitivity in the third trimester of pregnancy in previously insulin-treated gestational diabetes women undergoing GCs treatment, at variance with previous evidence [3]. Indeed, the likelihood of insulin supplementation in this limited retrospective cohort of pregnant women was significantly lower by 30% in those treated with αMD compared to their untreated counterparts. This observation, however, was not extended to women with gestational diabetes previously unexposed to exogenous insulin and in those with a de novo glycemic derangement needing insulin in the days following GCs.

Alpha-2 adrenoceptors are important regulators of blood glucose homeostasis since they inhibit insulin secretion from pancreatic beta-cells [17]. Impaired fasting glycemia and glucose-stimulated insulin secretion have been associated with overexpression of α2-adrenoceptors [18]. Although the administration of alpha2-adrenoceptor agonists generally increases blood glucose levels, the accompanying sympatholytic effects cannot be disregarded [19]. Despite the fact that the effect of alpha-adrenergic agonism on glucagon release remains undetermined, glucagon release may be mediated by alpha-2 adrenoceptors in rats [20].

The mechanism of action of αMD (alpha-2 adrenergic agonist) either at the central nervous system or at the periphery is largely undefined [21]. The study of the effect of αMD in patients with type 1 diabetes is currently ongoing [22,23], with evidence of amelioration of glycemic control in these patients. It has been observed that αMD may reduce beta-cell function in recent-onset type 1 diabetes mellitus patients. Several pathophysiological hypotheses have been postulated, including the inhibition of HLA-DQ8 presentation and the reduction of insulin-specific CD4+ T cell response in the peripheral circulation [22]. In women with type 1 diabetes mellitus and microalbuminuria or diabetic nephropathy, early BP-lowering treatment, including αMD, was associated with uncomplicated pregnancy outcomes [23].

Our observations may result from the insulin-induced changes in αMD uptake and metabolism, parallel with the dopamine uptake in brain cells by the dopamine transporter increased by insulin [24]. The adrenoreceptor pathway may be investigated as an alternative to beta-cell pharmacodynamics.

We acknowledge some limitations of the study. First, the sample size was quite limited, and the power of any significant association was rather reduced. Second, the retrospective collection and data analysis might have introduced investigator bias, and unmeasured confounders such as parity, blood pressure levels, and type of obstetric complications may have altered our unadjusted results. Although the retrospective evaluation of clinical data cannot demonstrate the effect of a drug on insulin action, especially when the sample size is limited, it may stimulate further refined research in the field with a more sophisticated study design. A future study may evaluate the need of insulin supplementation between two groups of pregnant women treated with betamethasone with gestational diabetes under insulin treatment with hypertensive disorders, including pre-eclampsia and women with gestational diabetes under insulin therapy without hypertensive disorders.

## 5. Conclusions

Our finding that the need of insulin supplementation in pregnant women under GCs treatment receiving αMD for gestational hypertension is reduced by 30% compared to women not receiving αMD may raise the hypothesis that αMD may ameliorate glucose levels in pregnancy by increasing insulin sensitivity.

## Figures and Tables

**Table 1 healthcare-10-00135-t001:** Insulin demand after administration of GCs in pregnancy.

	Group 1: GDM Already Treated with Insulin		Group 2: GDM on Medical Nutrition Therapy		Group 3: Without GDM	
	Increase in Insulin Dose		Insulin Treatment Initiation		Insulin Treatment Initiation	
	<30%/day	≥30%/day	No	Yes	No	Yes
No αMD Rx	13	37	12	35	9	29
Rx αMD	5	4	1	5	0	2
Difference between treatment with and without αMD	−30.0%		−8.5%		−23.7%	
Chi Square (*p*-value)	3.18 (0.035)		0.20 (0.32)		0.60 (0.22)	

GCs: glucocorticoids; GDM: gestational diabetes mellitus; Rx αMD: alpha-methyl-DOPA treatment.

## Data Availability

The data of the study can be obtained at https://figshare.com/ndownloader/files/32837282.

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
