# Peer review of "Alpha-Methyldopa May Attenuate Insulin Demand in Women with Gestational Diabetes Treated with Betamethasone"

_healthcare, 2022, doi:10.3390/healthcare10010135_

Round 1

Reviewer 1 Report

Thank you for the opportunity to review your manuscript.

The hypothesis and study design were simple yet well thought out. 

The results and discussion are also useful for future studies. 

I can also see statistical significance ( P < 0.05) with group 1 - this should be reflected in the abstract and discussion. 

High level publications will include statistically significant results in their abstract, discussion and conclusion.

Eg for the abstract:

The results showed that in the group of women with GDM under insulin treatment, who received αMD for hypertension, the increase in insulin needs was relatively lower (50 patients vs 9 patients, p=0.035) compared to all of the remaining women in the same group.

Something similar in the discussion and conclusion would be useful. Having a percentage is too crude for small numbers

Author Response

Reviewer 1 comments

  1. Thank you for the opportunity to review your manuscript

Reply: We thank the Reviewer for the constructive comments on our manuscript.

  1. The hypothesis and study design were simple yet well thought out

Reply: We agree, we kept the study design as simple as possible.

  1. The results and discussion are also useful for future studies

Reply: We thank the reviewer for this positive comment.

  1. I can also see statistical significance (P<0.05) with group 1 - this should be reflected in the abstract and discussion

Reply: According to reviewer’s remark, we added the relevant P-value in the revised abstract and revised results section of the manuscript. We also provided this significant effect in the first paragraph of the revised discussion section of the manuscript.

  1. High level publications will include statistically significant results in their abstract, discussion and conclusion

Reply: We agree and we modified the relevant sections accordingly. We also revised the conclusion section in line to reviewer’s suggestion.

  1. Eg for the abstract: The results showed that in the group of women with GDM under insulin treatment, who received αMD for hypertension, the increase in insulin needs was relatively lower (50 patients vs 9 patients, p=0.035) compared to all of the remaining women in the same group.

Reply: We revised the abstract in line with your proposal.

  1. Something similar in the discussion and conclusion would be useful. Having a percentage is too crude for small numbers

Reply: We added this part in the first paragraph of the discussion section. Although in this case we used a percentage, we also in the same phrase acknowledged the limited number of our retrospective cohort.

Reviewer 2 Report

A retrospective study is not the most adequate to demonstrate the effect of a drug on insulin action with only clinical data.

This is even more so when the number of hypertensive pregnant women treated with alpha methyldopa is low.

It is better to perform a classic retrospective cohort study with 2 arms, gestational diabetes in insulin therapy with pre-eclampsia and gestational diabetes in insulin therapy without pre-eclampsia, both treated with betamethasone, resulting in units of insulin / kg of weight.

The effect of alpha methyl dopa on insulin sensitivity is polemic. This research design is not the most adequate to clinically identify insulin demands in women with gestational diabetes and hypertension, as demonstrated by post hoc analysis

Author Response

Reviewer 2 comments

  1. A retrospective study is not the most adequate to demonstrate the effect of a drug on insulin action with only clinical data.

Reply: We strongly agree with your comment. We therefore, added an additional phrase in the limitation section to reflect your relevant comment.

  1. This is even more so when the number of hypertensive pregnant women treated with alpha methyldopa is low.

Reply: Our additional phrase in the revised limitation section underlines once again the limited number of participants.

  1. It is better to perform a classic retrospective cohort study with 2 arms, gestational diabetes in insulin therapy with pre-eclampsia and gestational diabetes in insulin therapy without pre-eclampsia, both treated with betamethasone, resulting in units of insulin / kg of weight.

Reply: We agree and we used reviewer’s reflection for future studies in the field in a phrase before the revised conclusion section.

  1. The effect of alpha methyl dopa on insulin sensitivity is polemic. This research design is not the most adequate to clinically identify insulin demands in women with gestational diabetes and hypertension, as demonstrated by post hoc analysis

Reply: We agree that the effect of alpha methyl DOPA on insulin sensitivity is rather controversial. For this reason, we tried to investigate this relation, in the rather difficult, however, field of gestational diabetes. Our goal was to bring our data forward, in order to shed light in this controversial deliberation in the literature.

Reviewer 3 Report

There is no reference to basic clinical parameters such as HbA1c.

The results are only drawn based on the need for exogenous inuslin - this is not enough.

Was only αMD used for treatment or were there also combination therapies among the examined patients?

Author Response

Reviewer 3 comments

  1. There is no reference to basic clinical parameters such as HbA1c.

Reply: We appreciate the reviewer’s comment. We believe that referring to insulin demand was the most descriptive way to present adequately the impact on glucose homeostasis. HbA1c, is used currently in non-pregnant subjects with diabetes and in pregnant women with pre-existing diabetes. In gestational diabetes, however, due to the transient nature of this entity, and to often concomitant anemia, the usefulness of HbA1c is rather limited. Most women with gestational diabetes have HbA1c lower than 6.0%. Lending credence to this, in the UK, the use of HbA1c in pregnancy is actually discouraged, as it is not considered to be a reliable index of glycemia in pregnancy by the National Institute for Clinical Excellence (NICE) (https://www.nice.org.uk/donotdo/do-not-use-hba1c-levels-routinely-to-assess-a-womans-blood-glucose-control-in-the-second-and-third-trimesters-of-pregnancy and https://www.guidelines.co.uk/diabetes/nice-diabetes-in-pregnancy-guideline/252595.article). 

  1. The results are only drawn based on the need for exogenous insulin - this is not enough.

Reply: We appreciate the reviewer’s comment. However, our work is based on clinical data regarding glycemia (and its concomitant management) after betamethasone administration. Women demonstrate significant hyperglycemia after being given antenatal betamethasone (Diabetes Res Clin Pract 2016; 118: 98-104). To the best of our knowledge, there is no other, non-invasive, way to quantify the effect in glycemic homeostasis, in the hospital environment, than insulin demand or need for insulin administration. For research purposes only, use of a glucose/insulin clamp could be implemented; such an approach was beyond the scope of the present study. 

  1. Was only αMD used for treatment or were there also combination therapies among the examined patients?

Reply: Because of the retrospective design we selected women with mild hypertension treated only with alpha methyl DOPA. After all, our goal was to investigate the relation between alpha methyl DOPA and glucose homeostasis. We included this important, though omitted, remark in the revised methodology section of the manuscript.

Round 2

Reviewer 3 Report

I have read the authors' comments and the revised version of the manuscript, and have no objections.